# Characterization of Biocompatibility of Functional Bioinks for 3D Bioprinting

**DOI:** 10.3390/bioengineering10040457

**Published:** 2023-04-09

**Authors:** Jinku Kim

**Affiliations:** Department of Biological and Chemical Engineering, Hongik University, Sejong 30016, Republic of Korea; jinkukim@hongik.ac.kr

**Keywords:** 3D bioprinting, bioink, biocompatibility, cell-material interaction, image analysis

## Abstract

Three-dimensional (3D) bioprinting with suitable bioinks has become a critical tool for fabricating 3D biomimetic complex structures mimicking physiological functions. While enormous efforts have been devoted to developing functional bioinks for 3D bioprinting, widely accepted bioinks have not yet been developed because they have to fulfill stringent requirements such as biocompatibility and printability simultaneously. To further advance our knowledge of the biocompatibility of bioinks, this review presents the evolving concept of the biocompatibility of bioinks and standardization efforts for biocompatibility characterization. This work also briefly reviews recent methodological advances in image analyses to characterize the biocompatibility of bioinks with regard to cell viability and cell-material interactions within 3D constructs. Finally, this review highlights a number of updated contemporary characterization technologies and future perspectives to further advance our understanding of the biocompatibility of functional bioinks for successful 3D bioprinting.

## 1. Introduction

Tissue engineering and regenerative medicine (TERM) is a promising interdisciplinary field that has been extensively explored as an attractive technology for ultimately restoring damaged or lost tissues or organs, thus improving the quality of life and eventually extending the human life span [1,2,3]. Therefore, by successfully repairing or regenerating organs and tissues damaged by disease or injury, it is expected to dramatically resolve existing chronic problems (lack of donated organs, various complications, etc.) [4]. However, despite extensive research endeavors over the past decades, tissue engineering strategies have not been successful in restoring the complex structure and functional three-dimensional (3D) tissues, such as the liver, due to the lack of suitable cell sources, biomaterials, and biomolecules [5,6,7].

Three-dimensional (3D) printing, which was first introduced in 1986, is a technique to produce complex 3D structures, and it can be used for various biomedical applications [8,9]. In particular, 3D bioprinting has been explored as a cutting-edge technology to create 3D customized constructs mimicking living tissues or organs such as bones, cartilage, or livers [10,11]. In the field of 3D bioprinting technology for manufacturing artificial 3D extracellular matrix (ECM), it is easy to control complex geometrical structures (e.g., craniofacial defects) and has excellent flexibility in designing fairly complex patient-specific scaffolds (e.g., desirable internal pores and pore interconnectivity) [12,13].

3D bioprinting utilizes additive manufacturing principles in which printing is carried out with appropriate biomaterial-based inks without incorporating cells or with suitable bioinks (cell-laden inks) to create 3D structures [14,15]. Currently available bioprinting methods can be categorized as extrusion-based [16], jetting-based (or droplet-based) [17], and vat photopolymerization-based (or light-based bioprinting) [18], according to ASTM standards. The basic principles, along with their advantages and limitations, have been published elsewhere [16,19,20]. Since the advent of 3D bioprinting, it has become extensively explored as a promising method to create functional organs or tissues because it allows to fabrication of precisely controlled functional organs, which have been desperately needed for end-stage treatment of diseased organs [21]. Besides organ printing, 3D bioprinting has been used to fabricate in vitro tissue models, drug screening, and biomedical devices [22,23,24]. For example, medical hook holders can be precisely manufactured from MED610 polymeric material by the PolyJet Matrix technology (PJM), one of the 3D bioprinting methods, which allows fabricating highly detailed and complex 3D models by using a matrix of printing heads to deposit tiny droplets of photopolymeric material onto a platform [25,26].

However, despite these remarkable developments in 3D bioprinting technology, there are still many challenges that need to be resolved [27]. One of the main challenges in the 3D bioprinting field has been to find an ideal bioink that is not only biocompatible with biological systems and harsh printing process conditions but can also provide printability with desired mechanical and biofunctional properties post-printing [28]. Over the past decade, researchers have made great endeavors searching for an ideal bioink for 3D bioprinting, consisting of biomaterials, living cells, and/or biomolecules, which possess desirable biocompatibility and printability as well (Figure 1a) [29,30,31]. A wide variety of natural or synthetic materials have been used as bioinks, and the development of advanced bioink materials and formulations for successful 3D bioprinting has been discussed in detail elsewhere [32,33,34].

The purpose of this review is to provide a better understanding of the ‘*biocompatibility*’ of biofunctional bioinks with regard to cell viability and cell-biomaterial interactions. In addition, this review presents recent advances in the development of in vitro analytical technologies, which can be used to characterize *biocompatibility*, such as cytotoxicity and biofunctionality of bioinks in 3D environments. Obviously, these characterization outcomes may provide much more representative and realistic information, which can be similarly observed in the clinical setting.

## 2. Characterization of Biocompatibility

### 2.1. Evolving Concept of ‘Biocompatibility’ of Bioinks

There is a noteworthy challenge in the development of ideal bioinks for 3D bioprinting with critical properties such as mechanical, biofunctional, and biocompatibility properties [36]. A bioink can be defined as a natural, synthetic, or hybrid material used in 3D bioprinting that is intended to interact with biological systems [37,38]. It typically consists of cells suspended in a hydrogel or other matrix material that can be printed by layer-by-layer deposition in additive manufacturing technology to fabricate the desired 3D structure [33,39]. Recently, Groll et al. defined bioink as “a formulation of cells suitable for processing by an automated biofabrication technology that may also contain biologically active components and biomaterials” [35]. Therefore, the bioink must provide suitable mechanical integrity and biological properties to maintain the bioink and cell integrity post-printing.

The *biofabrication window* paradigm has frequently been used to design bioinks for 3D bioprinting, which describes the compromise between printability and biocompatibility within 3D printed constructs (Figure 1b) [40,41]. In general, polymeric bioinks with higher crosslinking density or stiffness often compromise cell viability and cellular functionality, such as migration, proliferation, and matrix deposition [42]. Therefore, traditional bioinks with acceptable shape fidelity or printability often lack cell viability as well as biofunctionality [43]. Toward an ideal bioink with significantly improved printability and biocompatibility simultaneously, emerging bioinks are now being designed through careful control of rheological properties (e.g., shear-thinning abilities) during the printing process as well as biodegradation and biofunctionalization properties (e.g., cell adhesion, migration, proliferation, differentiation) [44,45].

There is a wealth of recent literature articles reviewing the printability for 3D bioprinting, which include bioprinting techniques, the concept of printability, and analysis of printability [40,46,47,48,49]. Therefore, this review focuses on the biocompatibility of bioinks, including the concept of biocompatibility and recent advances in analysis techniques to determine the biocompatibility of bioinks for 3D bioprinting.

The understanding and measurement of the biocompatibility of bioinks are unique to the design and development of an ideal bioink. The ultimate goal of biocompatibility testing is to ensure that the bioink is safe and effective for use in a specific application and to minimize the risk of adverse effects on biological function or structure. Unfortunately, we do not have a precise definition of biocompatibility and, subsequently, accurate measurements of biocompatibility. The term biocompatibility may refer to the property of a bioink to interact with recipient tissues and physiological systems treated with the bioinks [50]. In other words, it is the capacity of a bioink to interact safely with biological systems without eliciting any adverse local or systemic reactions. These include interactions with cells, tissues, and the immune system. For example, in the case of an implantable bioprinted 3D construct fabricated by bioinks, biocompatibility can influence the long-term performance of the construct and reduce the likelihood of rejection. In tissue engineering/regenerative medicine (TERM), biocompatibility is essential for promoting cell growth and tissue regeneration. The biocompatibility in 3D bioprinting technologies includes the expectation of active and controllable biofunctional contributions to the 3D construct. These may include interactions with endogenous tissues and/or the immune system, supporting favorable cellular activities (e.g., cell adhesion, growth, differentiation), facilitating molecular or mechanical signaling systems (mechanotransduction), and resisting infection and inflammation, all of which are essential for successful transplantation and function. With the advent of tissue engineering/regenerative medicine, the current concept of biocompatibility is evolving from a traditional goal for bio-safety (i.e., able to coexist with the endogenous tissue without eliciting any adverse local or systemic effects) to biofunctionality (i.e., able to actively promote desirable functions in the host. Therefore, we may directly apply the biocompatibility definition by Williams to the biocompatibility of a bioink [51,52];

“The Biocompatibility of a bioink for 3D bioprinting refers to the ability to perform its desired function that will support the appropriate cellular activity, including cell viability, adhesion, proliferation, and differentiation, in order to facilitate tissue regeneration, without eliciting any undesirable local or system effects in the eventual host”.

As the current concept of biocompatibility may integrate and promote the biofunctionality (i.e., able to promote desired function with respect to an appropriate host response) of an ideal bioink, which must possess enhanced cell viability, functionality, and printability simultaneously (Figure 1b). However, one of the most daunting challenges in the development of bioinks is to validate their biocompatibility. Moreover, if a material in the bioink is biodegradable, the degradation products behave likewise [53]. Furthermore, an important consideration in utilizing in vitro tests to determine the biocompatibility of bioinks is the additional components within a bioink. The majority of cell-laden bioinks are comprised of degradable or non-resorbable polymers such as poly(ethylene glycol) (PEG) hydrogels [54]. However, additional components other than polymeric materials may be incorporated into the bioink product to obtain the desired performance outcomes, such as printability [44]. These components may include fillers, plasticizers, and, notably, crosslinking agents, some of which may not be cell-friendly. Thus, leachables or surface-layered materials may impact the eventual clinical outcomes between the implanted bioink and the recipient (i.e., the patient).

The process by which a *bioink* may be determined to be *biocompatible* begins with a series of standardized in vitro assays and progresses to a stringent set of standardized in vivo tests as well as clinical trials. These tests can include cell viability assays, biofunctionality studies, including the characterization of cell-material interactions, animal studies, and human clinical trials. The results of these tests can provide information about the potential for adverse reactions, such as inflammation, infection, cellular responses to the matrices, and tissue damage. Therefore, biocompatibility is a critical aspect of bioink design and development, as it determines the ability of a bioink to interact safely with biological systems in a specific application.

Traditionally, the characterization of cell–material interactions using innovative techniques is most commonly performed on 2D cell culture environments such as cell culture on biomaterial surfaces [55,56]. However, these outcomes often inaccurately represent the complex interactions when cells are encapsulated within 3D matrices, which can be considered as “combination products” (cell scaffold) [57,58]. As compared to conventional 2D cell culture, this 3D microenvironment provides a more natural and realistic structure that better mimics the physicochemical cues of the ECM in vivo [59,60]. Although it is a daunting challenge to develop appropriately in vitro as well as in vivo testing to characterize cell-material interactions within 3D structures, the outcome would provide much more realistic information on the biofunctionality of 3D structures such as cell-laden bioinks [61]. The 3D encapsulation of cells within bioinks represents an increasingly complex technique for cell culture but permits the fabrication of constructs that further replicate the biomimetic cellular architecture of tissue engineering applications, leading to enhanced outcomes for the patient [62]. In designing new bioinks for extrusion bioprinting, initial cell screenings continue to be an established method to determine cell–material interactions. Thus, it is of crucial importance to determine cell–matrix interactions as well as the deposition of nascent ECM proteins using various advanced techniques.

Standardization for any process is laborious and time-consuming. It requires a lot of research and planning to properly standardize complicated processes, such as the characterization of the biocompatibility of bioinks, which may contain living cells. Although cell-laden bioinks has not been available for clinical studies until now, acellular 3D printed products, such as patient-specific surgical implants, are currently being used in clinic [63]. The best starting point for understanding the standardization of biocompatibility requirements is the International Organization for Standards (ISO), specifically Standard 10993, Biological Evaluation of Medical Devices. The ISO 10993 is the most extensive and standardized document available and is intended to provide international standards and methodologies to evaluate biomedical materials and devices. In particular, Part 5 describes tests for cytotoxicity, including detailed in vitro methods.

According to the ISO 10993 Part 5 (tests for in vitro cytotoxicity), there are three primary in vitro cellular toxicity assays that are rapid, sensitive, and inexpensive tests; (1) the agar diffusion test, (2) the direct contact test and (3) the elution, or extract dilution assay. Despite the paradigm shift of cell culture techniques from 2D to 3D to closely mimic the 3D in vivo environment [64,65], these tests can be directly applied to validate cell viability in 3D bioprinted constructs. Each method has its own advantages and limitations, and the choice of a method depends primarily on the type of cells and the experimental conditions. The agar diffusion, direct contact, and elution or extract dilution assays are based on changes in cellular morphology. The outcome from the cytotoxicity test can be assessed by changes in viable cell populations. The differences in the assays are dependent on how the test material is exposed to the cells.

The United States Food and Drug Administration (FDA) has substantially adopted the ISO guideline for medical device testing. In addition, the agency recently issued a guidance document, “Technical Considerations for Additive Manufactured Medical Devices” Guidance for Industry and Food and Drug Administration Staff (FDA-2016-D-1210), which outlines the requirements for the design, development, and validation of medical devices using additive manufacturing such as 3D bioprinting. Furthermore, the ISO has developed the ISO/ASTM 52900:2015 (ASTM F2792) Additive Manufacturing—General Principles—Terminology (2015), which provides a comprehensive overview of 3D bioprinting and its use in medical applications. Both documents provide detailed guidance on topics such as device design, manufacturing process, testing, and characterization for final devices fabricated by additive manufacturing.

### 2.2. Cell Viability

The primary purpose of a material biocompatibility assessment is to protect patient safety. From the bench-to-bedside process of biomaterials development, safety and efficacy must be emphasized. A bioink can be defined as any natural or synthetic materials used in bioprinting that are intended to interact with a biological system [37]. Cytotoxicity or cell viability assays have historically been used as the initial screening method to verify the biocompatibility of a biomaterial that may be used as a bioink or its component. Other methods are available to determine cytotoxic responses within the 3D structures. One of the most widely used testing methods to assess the cell viability of bioinks is MTT [3-(4,5-dimethylthiazol-2-tl)-2,5-diphenyltetrazolium bromide] assay, which is also connected to ISO 10993, Annex C. It is based on an optical density in the formazan crystals produced by viable cells and then quantified by spectrophotometry or image analysis [66,67]. Similarly, MTS [3-(4,5-dimethyl-3-yl)-5-(3-carboxymethoxyphenyl)-2-(4-sulfophenyl)-2H-tetrazolium)] assay has been developed and has become an increasingly popular method due to the elimination of the solubilization step to liberate the formazan crystals from the cells [68].

In addition to the standardized methods, AlamarBlue is another metabolic activity indicator that utilizes an oxidation-reduction system that changes color in response to the chemical reduction of the reagent in the growth medium during cell growth [69]. Additional cytotoxicity assays to monitor cell viability within 3D constructs include lactate dehydrogenase (LDH) [70], Trypan Blue (TB) [71], and live/dead assay [72]. In particular, the live/dead assay involves staining cells with a mixture of dyes, one (calcein-AM) of which stains viable cells while the other stains (ethidium homodimers) non-viable cells stained cells. The cells can then be visualized, and the percentage of viable cells can be calculated (Figure 2) [73]. Initially, procedures for assessing the cytotoxicity of bioinks were based on morphological changes leading to cell death, and they were utilized by many laboratories. While the distinction of good or bad cytotoxicity of bioinks seems rather intuitive and the quantitative definition of the cytotoxicity lacks consensus, and in fact, it depends on a number of factors, such as the chemical and physical nature of the biomaterial components and the duration of that exposure. According to ISO 10993, a reduction in cell viability by more than 30% is considered a cytotoxic effect. However, we must be mindful that the cytotoxicity tests of bioinks are performed within a 3D environment where the cells are embedded within a biomaterial. Thus, the outcomes may not be the same as those from the 2D setting. For example, the size and shape of a biomaterial could interfere with the penetration of molecules through 3D structures, creating a concentration gradient between the periphery and the center of the biomaterial. Diffusion and penetration limits for nutrients, oxygen, and reagents (e.g., fluorescent probes) must also be considered [74]. In addition, light scattering and autofluorescence of biomaterials may influence the outcomes of cytotoxicity of 3D constructs [75].

The majority of published work regarding the biocompatibility of bioinks used in 3D bioprinting has mainly focused on the cytotoxicity or cell viability of the inks [39]. The process parameters of 3D bioprinting significantly impact cell viability by providing shear stress, heat, radiation, mechanical impact, etc. [76]. Mitigation strategies for the improvement of cell viability have been proposed. For example, Ng et al. discussed the effects of the droplet impact velocity and volume on cell viability, suggesting that increasing cell density in cell-laden droplets improved cell viability as well as the proliferation of 3D-printed human primary cells [77].

### 2.3. Cell-Biomaterial Interactions

#### 2.3.1. Microscopy

Numerous techniques have recently been developed to characterize cells embedded in 3D structures, in which each method has its own advantages and limitations. Despite the difficulties (e.g., temporal and spatial resolution, high scattering, penetration depth) of 3D biological samples by their nature [78], advanced microscopy techniques can visualize those biological specimens at a high temporal and spatial resolution [79]. One of the most extensively used microscopy-based imaging techniques to capture cells in 3D structures is light microscopy (LM) or electron microscopy (EM). For example, cells cultured on biomaterials can be visualized by staining cell nuclei using either a blue fluorophore or DAPI and counterstaining the cell cytoskeleton (i.e., β-actin) by a green fluorophore or FITC [80]. In this way, cell morphology may be captured, as well as cell spreading, which is a crucial precursor to cell proliferation. Recently, more innovative microscopy-based technologies combined with image analysis have been extensively explored to better understand cell-biomaterial interactions [81].

#### 2.3.2. Laser Confocal Scanning Microscopy (LCSM)

Laser confocal scanning microscopy (LCSM) can acquire multiple 2D images at different depths and reconstruct 3D images by the process known as optical sectioning in a sample, generating high-resolution 3D images (Figure 3) [82]. This technique has been widely used in material science and life science, including tissue engineering and regenerative medicine [83,84,85]. With selective fluorescent labeling chosen by a researcher, the LCSM can provide precisely focused 3D images of subcellular structures such as nucleic acids, membranes, and mitochondria [59,86]. This method has been an extremely powerful tool to acquire 3D images of cell viability, proliferation, and differentiation of the cells grown inside 3D structures, including bioprinted constructs, despite the use of toxic labeling dyes such as calcein, which is routinely used for live/dead staining [87,88,89]. For example, the effects of cell density of adult stem cells cultured within 3D structures on cellular biocompatibility (viability, proliferation, and functionality) were examined by capturing 3D cell images within the printed 3D structures or on the surface of the structure by LCSM after staining the cells with fluorescence dyes (Figure 4) [90].

However, there are a number of limitations to the wider use of LCSM. For instance, the use of expensive antibodies for sample preparation and potentially toxic fluorescent dyes for scanning acquisition may have adverse effects on cell viability embedded in 3D structures. In addition, the phototoxicity from the use of high-energy lasers may further damage the biological samples. The introduction of multiphoton LCSM (e.g., two-photon LCSM) may reduce phototoxicity and photobleaching due to the use of low-energy photons [91]. The depth of field limit (~300 μm) could also restrict in-depth investigation of cell behavior inside 3D constructs [92].

#### 2.3.3. Two-Photon Laser Scanning Microscopy (TPLSM)

Two-photon laser scanning microscopy (TPLSM) is a form of fluorescence microscopy that utilizes the nonlinear process of two-photon absorption to excite fluorescent molecules. In this method, laser light is focused on the sample, and two photons are absorbed simultaneously by a fluorescent molecule, which then emits fluorescence [93]. The advantage of two-photon laser scanning microscopy over conventional one-photon fluorescence microscopy is that it allows much larger penetration depth into the specimens, as the longer wavelength of the laser light used (e.g., near-infrared) minimizes absorption and scattering by the biological samples. This makes it particularly useful for imaging deep inside live cells and tissues and other biological samples with 0.64 μm lateral and 3.35 μm axial spatial resolution [94]. This technique may be a superior alternative to confocal laser scanning microscopy (CLSM) due to the increased penetration depth, efficient light detection, and reduced photobleaching without compromising 3D resolution [95]. This method has recently been explored to elucidate cell-matrix interactions within 3D cell-laden bioinks. For example, Campos et al. visualized 3D cell spreading and distribution of MSCs inside 3D bioprinted constructs using TPLSM. They determined 3D cell morphology, volume, spatial distribution, and osteogenic and adipogenic differentiation of MSCs encapsulated in 3D bioprinted hydrogel constructs (Figure 5) [96].

#### 2.3.4. Scanning Electron Microscopy (SEM)

Scanning electron microscopy (SEM) has been widely used for imaging cells on the surface of biomaterials. Traditional SEM requires the cells to be fixed and dehydrated using ascending grades of ethanol and sputter coated with chromium, gold, or carbon, and it has been employed to assess material characterization (e.g., surface topography, nanoparticle morphology) [97,98]. It is also exploited to visualize images of the morphology of 3D-printed structures alone or cell-laden structures [99]. Recently, environmental SEM (ESEM) or cryo-SEM have been used for monitoring cellular activities in 3D printed bioinks such as hydrogels without essential water loss since conventional SEM usually requires low vacuum to obtain images of the cell-laden bioink structures unless the microscope has a special unit to freeze-dry the samples.

Although SEM is a powerful tool for acquiring images of both materials and biological samples due to its user-friendly interface, rapid image production, and ease of use, a number of limitations of SEM technologies include the limited size of the samples that can fit in the microscope chamber, the analyses of wet samples with conventional SEM [81]. Recent innovations in SEM technologies to generate high-resolution 3D images of cellular structures may be used to obtain the 3D microscopic images of cell-laden 3D printed structures for acquiring more details of cellular activities within 3D structures. For example, Hasegawa et al. used the focused ion beam SEM (FIB-SEM) to generate the images of cartilaginous fibrils and osteoblastic cytoplasmic processes, and they found that osteoblasts not only extend their cytoplasmic processes to the bone matrix but also stack these cell processes on the osteoid of the primary trabeculae [100]. Such recent progresses in SEM technologies may be directly applied to characterize the internal structure of 3D bioprinted constructs, including cell distribution and tissue development. This information can be useful to assess the structural quality of bioinks and biofunctionality of 3D bioprinted constructs, providing valuable insights into their potential for use in TERM applications.

#### 2.3.5. Atomic Force Microscopy (AFM)

Atomic Force Microscopy (AFM) is a member of scanning probe microscopy (SPM) technologies and is a powerful technology that has also been used to assess the topological investigation of surfaces of biomaterials at nano- and sub-nanometric resolution. Thus, the technology provides quantitative information on the relationship between the physicochemical properties of biomaterials and biological responses [101,102]. AFM probe techniques involve the quantification of how strongly a cell can adhere to the surface of a biomaterial. In cell biology, AFM can be used to attempt to distinguish cancer cells and normal cells based on the hardness of cells and to evaluate interactions between a specific cell and its neighboring cells in a competitive culture system, providing detailed biophysical information of cell–cell adhesion and cell–matrix adhesion forces [103].

AFM technique has become a rapidly emerging standard technique to collect more detailed information about surface and interface properties as compared with other microscopic techniques, especially in image analysis of biological samples. This is mainly due to the higher resolution measurements of biophysical data, such as adhesive forces in different materials and 3D surface profile acquisition with high atomic scale resolution at the z-axis (~1 Å). Moreover, unlike other microscopic technologies, AFM does not need complex sample preparation procedures (i.e., fixation and cryo-preparation methods) [104]. AFM is also gaining increasing interest as a useful tool for scaffold design, providing the opportunity to evaluate the mechanical properties of a wide range of biological samples on biomaterials under physiological conditions [105,106]. However, AFM has a limited scanning area (150 × 150 μm and a maximum height on the order of 10~20 μm) and the scanning speed, requiring several minutes for a typical scan, which often leads to thermal drift in the image, compared with other microscopic technologies such as SEM. Continuous innovations in AFM technology suggest a significant increase in the scanning speed and a decrease in image distortions induced by thermal drift [107,108]. In the case of cell-laden bioink systems, fully encapsulated cells in the bioinks may not come in contact with the AFM probe. Thus, non-contact AFM may be useful for obtaining information regarding cell-matrix interactions without the destruction of the printed construct [109].

#### 2.3.6. Traction Force Microscopy (TFM)

Another useful microscopic technique to measure the mechanical properties of cells is traction force microscopy (TFM). In anchorage-dependent cells, the traction stresses (force/area) generated between cell adhesion and the extracellular matrix (ECM) modulate various cellular biofunctionality such as cell spreading and migration, proliferation, and differentiation [110,111,112,113]. The TFM has been used to determine the traction forces between cells and ECM [114]. The method works by applying a small force on cells or tissues and by measuring the resulting deformation, thereby calculating the traction forces that the cells or tissues generate on the substrate they are grown on. This information can provide a better understating of complex mechanical interactions between cells and surrounding microenvironments (e.g., mechanotransduction) [115,116].

Conventional TFM measures only shear traction forces (parallel to the plane of the substrate). However, recent studies have revealed that cells on the substrate can generate significant vertical (normal) tractions, implying that cellular traction force generation is more complex than initially thought [117,118]. For example, Legant et al. demonstrated that both shear and normal tractions moved with the extending leading edge. This dynamic colocalization of force distributions at the cell periphery was also found during cell spreading (Figure 6). These forces produce significant rotational moments about focal adhesions in both protruding and retracting peripheral regions. Temporal multidimensional traction force microscopy analysis of migrating and spreading cells shows that these rotational moments are highly dynamic, propagating outward with the leading edge of the cell [119]. In order to obtain the mechanotransduction information in cell-laden bioink using 3D TFM, fluorescent beads need to be co-encapsulated with the cells, which may unintentionally modify the bioink’s rheological properties. With this regard, transparent samples may be preferred due to the advantage of clearly visualizing the fluorescent beads.

One of the disadvantages of TFM is limited spatial resolution, which is above one micron, preventing the detailed mechanotransduction mechanism [120]. A recent study suggests that substrate surface modification may be able to enhance the spatial resolution of the TFM, comparable to fluorescence microscopy, to characterize macromolecular scale traction events [121].

#### 2.3.7. Optical Coherence Tomography (OCT)

Optical coherence tomography (OCT) is a promising non-invasive imaging technology that uses light to produce detailed, high-resolution cross-sectional images of biological tissues. The technology is based on the principles of low-coherence interferometry and is similar to ultrasound imaging in terms of the information it provides, but it uses light instead of sound waves to image the tissue [122]. OCT depends on the scattering property of the samples rather than on fluorescence or ionizing radiation, which poses a low risk of altering or changing the samples being examined [123]. OCT is a promising tool that is able to provide real-time, detailed information on the 3D constructs’ structure, cellular dynamics, and tissue development of biological or engineered tissues with high resolution (1–10 μm) and deep penetration (1–5 mm) [124,125,126,127]. A study has demonstrated that OCT may be an effective means of producing 3D visualization of cellular activities (e.g., chemotaxis) within 3D hydrogel constructs [128]. Wang et al. recently reported that OCT might be a key tool for non-invasive and quantitative characterization, design optimization, and fabrication refinement of 3D bioprinted hydrogel scaffolds [129,130]. Therefore, this exciting technology may produce detailed information on the quantitative relationship between 3D bioprinted cell-laden structures and biological outcomes.

#### 2.3.8. Micro Computed Topography (MicroCT)

Microcomputed tomography (microCT) is one of the most widely used non-destructive techniques in the field of TERM, which uses X-rays to produce 3D images of small, high-resolution objects. In microCT technology, the sample is placed on a rotating stage, and X-rays passing through the sample from different angles are detected; then, the collected radiographic projections are used to reconstruct a 3D image of the sample’s structure [131]. The resulting image provides detailed information about the internal and external structure of the sample, including its shape, density, and composition. Micro-CT has been extensively used in many applications, including biology, materials science, engineering, and medicine [132]. In particular, the method is widely used in the characterization of the mineralization process within 3D scaffolds based on in vitro culture conditions, and osteogenic properties from 3D quantification of new bone formation ex vivo as well as in vivo pre-clinical models [133,134,135].

While microCT is commonly used to examine morphological characteristics of hard tissues or various materials such as metals, ceramics, or some polymers, the method has also been explored to examine soft matters such as hydrogels, cells, and tissues with the incorporation of heavy elements such as osmium tetroxide (OsO4) [136,137,138]. However, the limitations of this technology include the nominal resolution in the range of 5~50 μm of image quality and difficulty in finding the correct threshold for materials or tissues with similar absorption coefficients [139]. In order to overcome such limitations, advanced microCT technologies have become available to assess the morphological information of soft tissues or materials. For example, phase contrast (PC) microCT uses X-ray refraction and phase shifting without phase contrast agents that are commonly used in X-ray absorption [140,141]. Now, it is possible to examine cell behavior within 3D cell-containing scaffolds based on cell labeling with a radio-opacifier such as barium sulfate [142].

To improve the resolution, synchrotron radiation micro-computed tomography (SRµCT) has been developed, which is based on the use of synchrotron X-rays (i.e., a parallel beam of X-rays with a very narrow bandwidth) [143]. These semi-coherent X-rays allow magnification optics to be used, resulting in a spatial resolution below 1 μm. For example, Thurner et al. examined the qualitative as well as quantitative differences in morphology, cell distribution, and cell adhesion between fibroblasts (human foreskin fibroblasts, HFF) and osteoblast-like cells (MC3T3) on 3D poly(ethylene terephthalate) (PET) multifilament yarns by using SRµCT (Figure 7) [144].

#### 2.3.9. Other Approaches

A number of innovative and compelling imaging technologies, in combination with computer-assisted imaging analyses, have become available to better understand 3D cell-biomaterial interactions. Besides the aforementioned advanced technologies, other methods have been continuously developed for quantitative analyses of cellular responses to 3D printed constructs. For example, the fluorescence resonance energy transfer (FRET) technique has been explored to quantify the relationship between cell adhesion ligand-receptor bonds and the phenotypic expression of cells embedded in 3D structures [145]. For example, a previous study demonstrated that the method could be useful in quantifying the direct relationship between the number of receptor bonds and cellular responses (e.g., proliferation and differentiation) of osteoblast-like cells (MC3T3) [146].

Another new technique to quantify cell-matrix interactions is multiple particle tracking microrheology (MPT). Using this technology, the thermal motion of the probe embedded in the hydrogel network can be measured and characterized spatiotemporal rheological properties in the pericellular region during cell-mediated remodeling [147]. MPT is a powerful technique to better understand the dynamics of cell-matrix interactions (e.g., remodeling dynamics of mesenchymal stem cells during cellular processes such as migration), advancing material designs that manipulate these processes for TERM applications [148].

Besides analytical technologies to better understand cellular responses in 3D embedded constructs, in silico predictive methods can be used to optimize cell viability as well as cell-matrix interactions of 3D printed bioinks with high accuracy [149]. For example, a data-driven predictive modeling approach based on machine learning was used to predict cell viability and the effects of process parameters such as UV intensity, exposure time, and bioink concentration on cell viability with high accuracy [150]. Therefore, we may be able to utilize in silico predictive models to better understand biocompatibility and perhaps to improve cell viability, cell-cell interactions, and cell-matrix interactions of 3D bioprinting in the near future.

## 3. Summary and Future Perspectives

In the field of tissue engineering and regenerative medicine (TERM), 3D bioprinting can be used to create functional tissue constructs that are used to repair or restore complex 3D defects in tissues or organs. At the same time, researchers are making continuous progress in producing ‘ideal’ bioinks for 3D bioprinting to revolutionize the field of TERM. Herein, this review briefly describes the evolving concept of the ‘*biocompatibility*’ of bioinks and advanced characterization technologies with respect to 3D image analyses to evaluate the biocompatibility, especially for investigating cell viability and cell-matrix interactions within 3D bioprinted constructs, which eventually determine the biological performance of bioinks after implantation in the body.

To date, there is no single and decisive assay to measure the biocompatibility of 3D bioprinted constructs. Thus, combinatorial approaches using multiple characterization methods have been used to investigate the biocompatibility of the bioinks for the acquisition of more detailed information. This emphasis is reinforced by the notion that all of the tests, either alone or in combination, continue to be of value when assessing the biocompatibility of a bioink material during the developmental phase and even into clinical applicability. Contemporary testing paradigms acknowledge that individual testing methods currently in place cannot absolutely screen in or out a material. Therefore, evolving bioinks testing strategies must continue to utilize a combinatorial approach as the technologies and methods for truly biocompatible bioinks for successful 3D bioprinting. For example, LCSM and SEM can be simultaneously used to examine the morphology of the cells and cellular function, such as migration and proliferation of the cells embedded in 3D bioprinted constructs [151]. Furthermore, a real-time, non-invasive, and non-destructive assay would be of crucial importance to assess the biocompatibility and perhaps to quantify in situ 3D cellular processes such as cell migration or even differentiation, matrix production throughout 3D porous constructs and overall biological performance of the construct [152,153].

In addition, there is an urgent need to develop high throughput in vitro assays to screen the cellular activities of cells within bioink constructs. Traditionally, in vitro 2D cell culture systems have been useful for screening the biocompatibility of biomaterials. However, it has become clear that cell-matrix interactions in 3D cell culture systems distinctively differ from those in traditional 2D substrates [110,154]. Moreover, the cells may respond to the natural tissue ECM, which is a highly dynamic structural environment, compared with less sophisticated 3D culture systems. Thus, high throughput 3D characterization platforms will be able to interrogate libraries of candidate bioinks for eventual patient applications effectively, efficiently, and in a standardized manner. There is no doubt that advanced and innovative imaging technologies would play pivotal roles in developing high throughput in vitro characterization systems to determine the biocompatibility of bioinks in 3D environments. LCSM technologies with high resolution and enhanced in-depth limit would also be very beneficial to assess 3D cell-biomaterial interactions within 3D structures with less toxic fluorescent dyes that can stain viable cells without compromising cell behavior. In fact, second-harmonic generation microscopy (SHGM) has been developed in combination with two-photon fluorescence microscopy to monitor cell-material interactions for higher depth and resolution with less phototoxicity and photobleaching [78,155]. Furthermore, with the advanced mass spectrometry and bioinformatics software development, proteomics may be considered as another high throughput technology to analyze the ‘biocompatibility’ of bioinks in terms of the relationship between adsorbed proteins and the biological responses when a bioink is implanted into the host [156,157]. Proteomics may provide valuable information on the adsorbed protein–bioink interactions, which can further help to identify the proteins that play a role in the biological responses, such as foreign body responses to the bioink material, responsible for its biocompatibility [158].

As an alternative approach to in vitro 3D bioprinting, in vivo bioprinting may be considered to resemble the realistic 3D environment in which the bioinks are directly deposited on or in a patient [159]. Recently, this approach has been explored to bioprint ear-like tissue constructs in muscle defects in mice without surgical implantation of the constructs [160]. In this case, high throughput in vivo characterization system may provide much more valuable information with regard to cell-matrix interactions of a candidate bioink in 3D in vivo environments, despite the rapid progress of the development of advanced analytical technologies with high resolution and throughput. Destructive and laborious assays such as histology and immunohistochemical assays may not be replaced due to excellent sensitivity and specificity. Finally, although these exciting novel techniques aforementioned may not be widely used in the near future in a standardized manner, it is believed that they should be able to produce more relevant and realistic outcomes in vivo as well as in a clinical setting one step further to an ‘ideal’ bioink for 3D bioprinting.

## Figures and Tables

**Figure 1 bioengineering-10-00457-f001:**
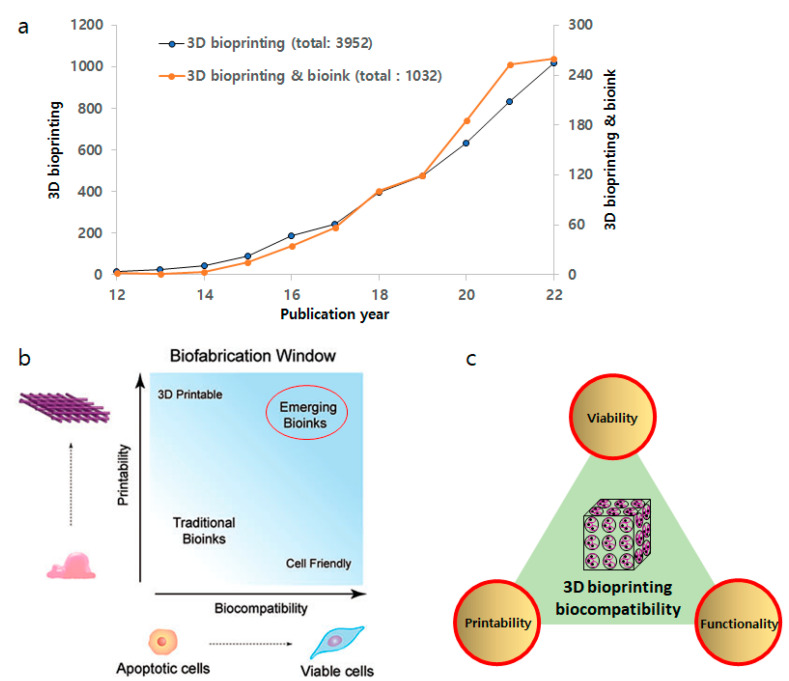
(**a**) Research growth in the field of 3D bioprinting and bioink. Data from Web of Science using “3D bioprinting” & “bioink”. (**b**) Biofabrication window. There has been a paradigm shift from traditional bioinks to emerging bioinks with better biocompatibility and printability. Adapted from ref [35]. (**c**) Integration of critical properties of bioinks.

**Figure 2 bioengineering-10-00457-f002:**
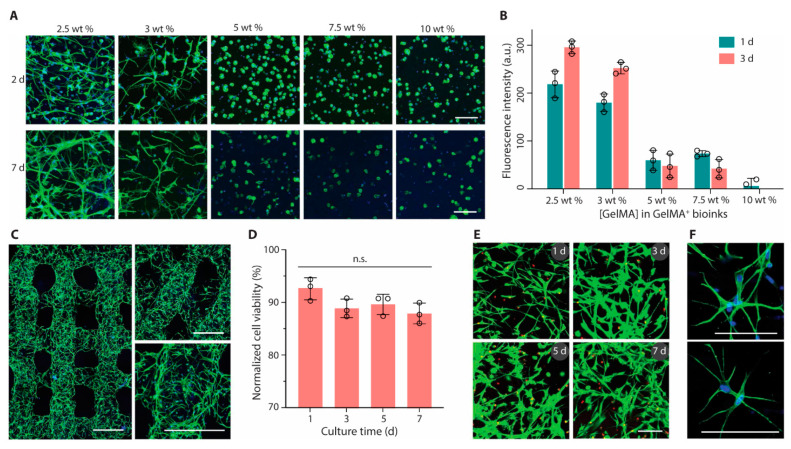
Bioprinting and culture of constructs containing astrocytes. (**A**) Fluorescence microscopy images (green, phalloidin; blue, DAPI) and (**B**) metabolic activity of astrocytes encapsulated in GelMA+ hydrogels with varied GelMA concentrations (2.5 to 10 wt%). (**C**) Confocal laser scanning microscopy images (green, phalloidin; blue, DAPI) of printed multilayer lattices with 90° or 60° angles between adjacent layers (2 days). (**D**) Cell viability and (**E**) Live/Dead-stained astrocytes (green, calcein AM; red, ethidium homodimer-1) after bioprinting and culture for up to 7 days. (**F**) High-magnification fluorescence microscopy images of astrocytes (green, phalloidin; blue, DAPI) in printed constructs (2 days). GelMA+ (2.5 wt%) and astrocytes were used for bioprinting in (**C**) to (**F**). Scale bars, 100 μm (**A**,**E**,**F**) and 500 μm (**C**). Reprinted with permission from ref [73].

**Figure 3 bioengineering-10-00457-f003:**
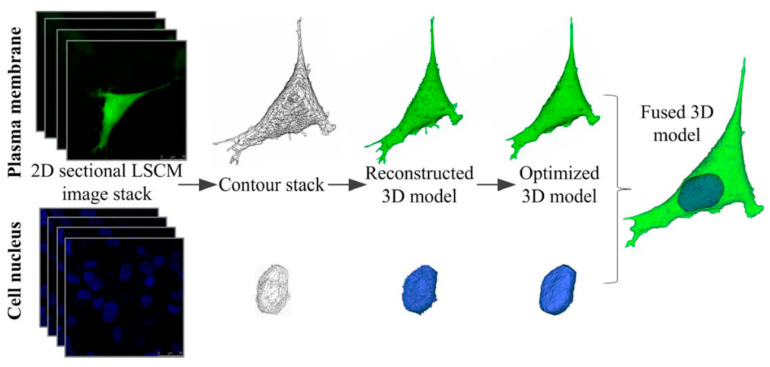
3D reconstruction procedure of cell model. The plasma membrane and cell nucleus were reconstructed separately and finally fused into a cell model (Green:GFP, Blue: DAPI). Reprinted with permission from ref. [82], Copyright 2013 John Wiley & Sons.

**Figure 4 bioengineering-10-00457-f004:**
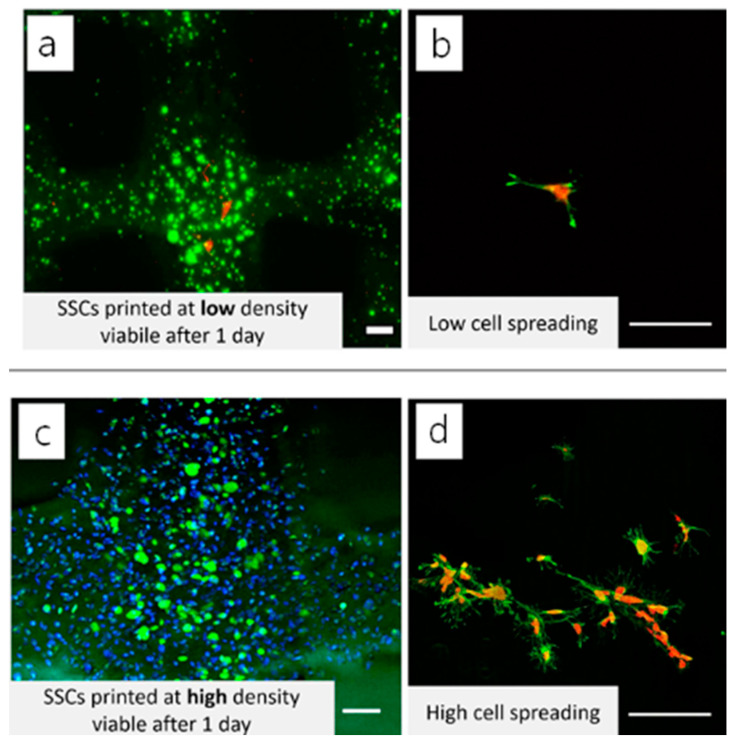
3D reconstructed LCSM image of cell viability of 3D printed GelMA structures. Adult stem cells were encapsulated in the 3D-printed GelMA at (**a**) low and (**c**) high cell densities. The cells were also seeded on the 3D printed structures at (**b**) low and (**d**) high cell densities, respectively. Adapted with permission from ref [90]. Copyright 2019 Elsevier.

**Figure 5 bioengineering-10-00457-f005:**
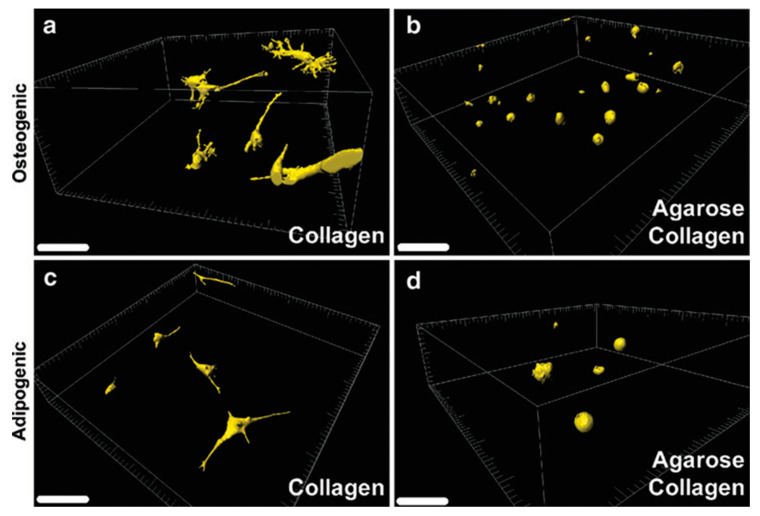
Assessment of cell spreading and distribution by TPLSM. MSC-laden collagen (**a**,**c**) and agarose–collagen (**b**,**d**) samples were cultured in osteogenic and adipogenic differentiation media, respectively. Scale bars represent 100 mm. Reprinted with permission from ref [96], Copyright 2014 Mary Ann Liebert.

**Figure 6 bioengineering-10-00457-f006:**
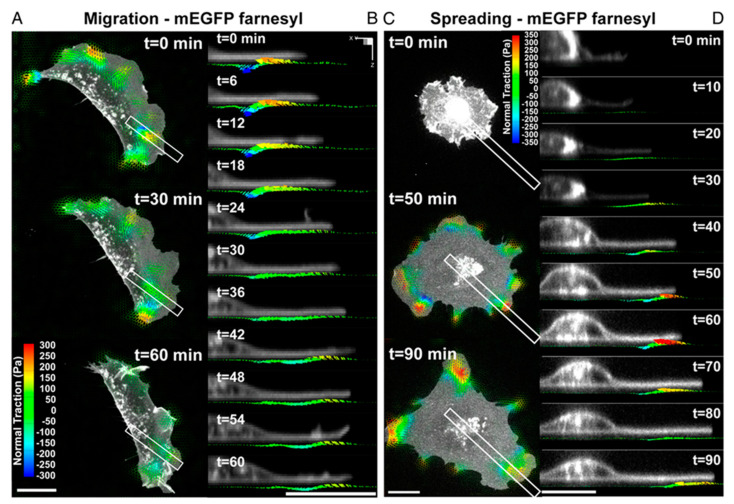
Dynamic measurements of 2.5D traction stress. (**A**) Time−lapse images depicting traction stress vectors color−coded by the normal component generated by a migrating MEF expressing mEGFP−farnesyl. As the cell moves (toward right), rotational moments are applied in the protruding front as well as the sides. (**B**) Time−lapse cross−sectional views of the inset outlined in A showing dynamic rotational moments that move with the thin protruding cellular body during cell migration. (**C**) Time−lapse images of mEGFP−farnesyl–expressing MEF undergoing spreading. No significant vertical traction stresses are detected until the cell extends thin protrusions and flattens against the substratum. Minimal tractions are detected under the nucleus. (**D**) Time−lapse cross-sectional views of the inset outlined in C. Comparable to migrating cells, rotating moments progress outward with the leading edge and remain localized to the cell periphery. (All scale bars, 20 μm) Reprinted with permission from ref [119].

**Figure 7 bioengineering-10-00457-f007:**
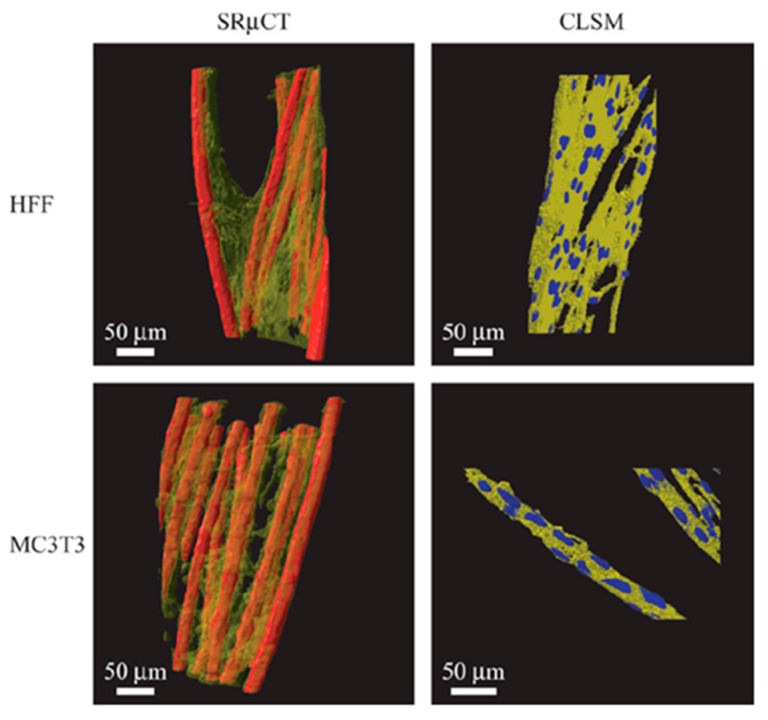
SRµCT 3D images of the two different cell cultures (transparent yellow) on the scaffold (red) compared to rendered confocal laser scanning microscopy (CLSM) image stacks of similar cell cultures. The SRµCT shows similar morphology as the CLSM images, validating the visualized morphologies even with coarser resolution than CLSM and further show qualitative and quantitative differences in cell distribution and adhesion by image analysis. The HFF cells are rather inhomogeneously distributed within the yarn, whereas MC3T3 cells are distributed homogeneously, lining with the individual filaments. As a result, the adhesion surface area of HFF cells is greatly reduced compared to the MC3T3 cells. Reprinted with permission from ref. [144], Copyright 2005 John Wiley & Sons.

## Data Availability

Not applicable.

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
