# Peer review of "Characterization of Biocompatibility of Functional Bioinks for 3D Bioprinting"

_bioengineering, 2023, doi:10.3390/bioengineering10040457_

Round 1

Reviewer 1 Report

Dear Author,

The article concerns an interesting field of application of 3D printing, but it requires many corrections.

1. In the text, the phrase we did is used, and only one author is mentioned in the article.

2. Articles currently do not meet the editorial in terms of structure.

3. Recommends including in the introduction the analysis of such technologies as PJM and medical materials:

a) Analysis of Metrological Quality and Mechanical Properties of Models Manufactured with Photo-Curing PolyJet Matrix Technology for Medical Applications, DOI10.3390/polym14030408

b) Tribological Properties of Medical Material (MED610) Used in 3D Printing PJM Technology, DOI10.17559/TV-20220111154304

4. The quality of the drawings should be improved.

5. There are syntax errors in the text, please read the entire publication and correct it.

6. Suggests shortening the descriptions under the drawings.

7. The work lacks a clear, large chapter describing in detail the known and used materials. In addition, the same chapter should apply to 3D printing technology in this area.

8. There is no analysis of databases, e.g. Web of Science, Scopus etc., in the field of conducted research.

regards,

Reviewer

Author Response

The article concerns an interesting field of application of 3D printing, but it requires many corrections.

  1. In the text, the phrase we did is used, and only one author is mentioned in the article.

Response: The author made corrections in the text as reviewer’s pointed out.  

  1. Articles currently do not meet the editorial in terms of structure.

Response: The author reorganized the structure of the article to meet the editorial.

  1. Recommends including in the introduction the analysis of such technologies as PJM and medical materials:
  2. a) Analysis of Metrological Quality and Mechanical Properties of Models Manufactured with Photo-Curing PolyJet Matrix Technology for Medical Applications, DOI10.3390/polym14030408
  3. b) Tribological Properties of Medical Material (MED610) Used in 3D Printing PJM Technology, DOI10.17559/TV-20220111154304

Response: The author included in the instruction the analysis of such technologies as PJM and medical materials as the reviewer recommended. (page 2, lines 67~71)

  1. The quality of the drawings should be improved.

Response: The drawings was significantly modified as the reviewer pointed out and the author believes that the quality of the drawings was greatly improved. (page 3, Figure 1)

  1. There are syntax errors in the text, please read the entire publication and correct it.

Response: The author carefully read the entire manuscript and made corrections of syntax errors throughout the text.

  1. Suggests shortening the descriptions under the drawings.

Response: The author shortened the descriptions under the drawings as suggested. (page 3, Figure 1)

  1. The work lacks a clear, large chapter describing in detail the known and used materials. In addition, the same chapter should apply to 3D printing technology in this area.

Response: The author agreed with the reviewer’s comment, but instead of adding large chapter to this review, detailing the materials and 3D printing technology, this review article focus on the characterization of the biocompatibility of bioink for 3D biopriting. The author included brief comments and added more recent references regarding recent development of materials for 3D printing as well as 3D printing technology. (page 2, lines 59~61 and lines 79~82)

  1. There is no analysis of databases, e.g. Web of Science, Scopus etc., in the field of conducted research.

Response: The author included the analysis of database (Web of Science) in the field of conducted research. (page 2, Figure 1)

Reviewer 2 Report

Comments:

1.       Biocompatibility should not only be constrained to bio-inks but instead a more holistic approach should be taken to evaluate biocompatibility in terms of the entire bioprinting process. Hence, the authors may consider revising the manuscript title – “Biocompatibility of 3D printing process”.

2.       The authors should consider revising the sentence “currently available bioprinting methods can be categorized extrusion-based, droplet-based and light-based”. According to ASTM standards, the different 3D bioprinting techniques should be categorized as extrusion-based, jetting-based and vat photopolymerization-based. The authors can refer to some of the following papers below:

a.       Extrusion

                                                               i.      "Current advances and future perspectives in extrusion-based bioprinting." Biomaterials 76 (2016): 321-343.

b.       Jetting

                                                               i.      "Inkjet bioprinting of biomaterials." Chemical Reviews 120, no. 19 (2020): 10793-10833.

c.       Vat photopolymerization  

                                                               i.      "Vat polymerization-based bioprinting—Process, materials, applications and regulatory challenges." Biofabrication 12, no. 2 (2020): 022001.

3.       The authors may want to use a more appropriate definition for biocompatibility.

a.       It should be linked to cell adhesion, viability, and proliferation etc in bioprinting context.

4.       Other than mentioning the different assay to measure cell viability, what are some of the recent work that have been done to improve the cell viability during 3D bioprinting process? The authors can refer to some of the following papers:

a.       "Controlling droplet impact velocity and droplet volume: Key factors to achieving high cell viability in sub-nanoliter droplet-based bioprinting." International Journal of Bioprinting 8, no. 1 (2022).

b.       "Prediction of cell viability in dynamic optical projection stereolithography-based bioprinting using machine learning." Journal of Intelligent Manufacturing (2022): 1-11.

c.       "Predicting and elucidating the post-printing behavior of 3D printed cancer cells in hydrogel structures by integrating in-vitro and in-silico experiments." Scientific Reports 13, no. 1 (2023): 1211.

5.       Similarly, what are some of the important parameters of biomaterials to improve the cell viability in those 3D structures for Section 2.3?

Author Response

Thank you for the valuable comments. The author revised the manuscript to improve the manuscript as recommended.

  1. Biocompatibility should not only be constrained to bio-inks but instead a more holistic approach should be taken to evaluate biocompatibility in terms of the entire bioprinting process. Hence, the authors may consider revising the manuscript title – “Biocompatibility of 3D printing process”.

Response: The author understands the reviewer’s point of view regarding the biocompatibility in general. But the author tried to briefly overview the characterization methods to evaluate the biocompatibility of 3D cell-laden bioinks for 3D printing using advanced analytical techniques, so the title may be appropriate to accommodate the contents of the manuscript.

  1. The authors should consider revising the sentence “currently available bioprinting methods can be categorized extrusion-based, droplet-based and light-based”. According to ASTM standards, the different 3D bioprinting techniques should be categorized as extrusion-based, jetting-based and vat photopolymerization-based. The authors can refer to some of the following papers below:
  2. Extrusion
  3. "Current advances and future perspectives in extrusion-based bioprinting." Biomaterials 76 (2016): 321-343.
  4. Jetting
  5. "Inkjet bioprinting of biomaterials." Chemical Reviews 120, no. 19 (2020): 10793-10833.
  6. Vat photopolymerization 
  7. "Vat polymerization-based bioprinting—Process, materials, applications and regulatory challenges." Biofabrication 12, no. 2 (2020): 022001.

 Response: The reviewer’s comment was well taken. The author revised the manuscript and included the references as recommended (pages 2, lines 59~61)

  1. The authors may want to use a more appropriate definition for biocompatibility.
  2. It should be linked to cell adhesion, viability, and proliferation etc in bioprinting context.

 Response: The author revised the definition of the biocompatibility as suggested. (page 4, lines 150~153)

  1. Other than mentioning the different assay to measure cell viability, what are some of the recent work that have been done to improve the cell viability during 3D bioprinting process? The authors can refer to some of the following papers:
  2. "Controlling droplet impact velocity and droplet volume: Key factors to achieving high cell viability in sub-nanoliter droplet-based bioprinting." International Journal of Bioprinting8, no. 1 (2022).
  3. "Prediction of cell viability in dynamic optical projection stereolithography-based bioprinting using machine learning." Journal of Intelligent Manufacturing(2022): 1-11.
  4. "Predicting and elucidating the post-printing behavior of 3D printed cancer cells in hydrogel structures by integrating in-vitro and in-silico experiments." Scientific Reports13, no. 1 (2023): 1211.

Response: The author added the sentences in the section to include recent work to improve the cell viability during 3D bioprinting processes as suggested. (page 7, lines 277~283)

  1. Similarly, what are some of the important parameters of biomaterials to improve the cell viability in those 3D structures for Section 2.3?

Response: Again, the author added the sentences in the section to accommodate the reviewer’s comment. (page 15, lines 527~535)

Round 2

Reviewer 1 Report

Dear Authors,

The article is improved and can be published in the presented form.

Regards,

Reviewer

Reviewer 2 Report

The author has addressed most of the comments, the paper can be accepted in present form